# Altered Cerebral Blood Flow in the Progression of Chronic Kidney Disease

**DOI:** 10.3390/jpm13010142

**Published:** 2023-01-11

**Authors:** Weizhao Lin, Mengchen Liu, Xixin Wu, Shandong Meng, Kanghui Yu, Huanhuan Su, Quanhai Liang, Feng Chen, Jincheng Li, Wenqin Xiao, Huangsheng Ling, Yunfan Wu, Guihua Jiang

**Affiliations:** 1The Department of Medical Imaging and Nuclear Medicine, Guangdong Medical University Zhanjiang Campus, Zhanjiang 524023, China; 2The Department of Medical Imaging, Guangdong Second Provincial General Hospital, Guangzhou 510000, China; 3The Department of Nephrology, Guangdong Second Provincial General Hospital, Guangzhou 510000, China; 4The Department of Renal Transplantation, Guangdong Second Provincial General Hospital, Guangzhou 510000, China

**Keywords:** chronic kidney disease, cerebral blood flow, cognitive function

## Abstract

Background: In chronic kidney disease (CKD), cognitive impairment is a definite complication. However, the mechanisms of how CKD leads to cognitive impairment are not clearly known. Methods: Cerebral blood flow (CBF) information was collected from 37 patients with CKD (18 in stage 3; 19 in stage 4) and 31 healthy controls (HCs). For CKD patients, we also obtained laboratory results as well as neuropsychological tests. We conducted brain perfusion imaging studies using arterial spin labeling and calculated the relationship between regional CBF changes and various clinical indicators and neuropsychological tests. We also generated receiver operator characteristic (ROC) curves to explore whether CBF value changes in certain brain regions can be used to identify CKD. Results: Compared with HCs, CBF decreased in the right insula and increased in the left hippocampus in the CKD4 group; through partial correlation analysis, we found that CBF in the right insula was negatively correlated with the number connection test A (NCT-A) (r = −0.544, *p* = 0.024); CBF in the left hippocampus was positively correlated with blood urea nitrogen (r = 0.649, *p* = 0.005) and negatively correlated with serum calcium level (r = −0.646, *p* = 0.005). By comparing the ROC curve area, it demonstrated that altered CBF values in the right insula (AUC = 0.861, *p* < 0.01) and left hippocampus (AUC = 0.862, *p* < 0.01) have a good ability to identify CKD. Conclusions: Our study found that CBF alterations in the left hippocampus and the right insula brain of adult patients with stage 4 CKD were correlated with disease severity or laboratory indicators. These findings provide further insight into the relationship between altered cerebral perfusion and cognitive impairment in patients with non-end-stage CKD as well as, additional information the underlying neuropathophysiological mechanisms.

## 1. Introduction

Chronic kidney disease (CKD) is defined as damage to the structure and function of the kidney caused by multiple causes that lasts for three months or longer. The incidence of CKD has been increasing rapidly for decades, and more than 850 million people worldwide are now living with this condition; CKD has become a growing public health issue [1,2]. Cognitive impairment is common in patients with end-stage renal disease (ESRD) and is related to its severity [3,4,5], specifically in orientation, attention, and executive ability [6]. ESRD combined with cognitive impairment can significantly increase the morbidity and mortality rate. However, the clinical manifestations of CKD combined with cognitive impairment are relatively insidious and easily overlooked by clinicians. Studies have found that compared with healthy controls, patients with CKD stage 3 or 4 have different rates of decline in functions such as language, attention, memory, and executive function [7,8,9]. However, the neuropathological mechanisms of non-end-stage CKD combined with cognitive dysfunction are not fully understood.

Three-dimensional pseudo-continuous ASL (3D-pCASL) technology is a noninvasive perfusion imaging technique used to quantify human cerebral blood flow (CBF) under various conditions. CBF is an important quantitative physiological indicator related to brain metabolism. ASL has been applied to the study of many brain diseases, such as subcortical vascular cognitive impairment [10], Alzheimer’s disease [11], and intracranial atherosclerotic disease [12,13]. In recent years, studies have been conducted using ASL to investigate the characteristics of CBF in CKD patients. Jiang et al. applied ASL to ESRD patients undergoing peritoneal dialysis and hemodialysis, ESRD patients not receiving dialysis, and healthy controls; they found that ESRD patients had increased average CBF compared to healthy controls and suggested that the degree of anemia may be a major risk factor for cognitive impairment [14]. Liu et al. used ASL-MRI to explore changes in CBF in children and young adults with CKD and showed that children and young adults with CKD stages 2–4 had higher global CBF than controls [15]. However, it is unclear whether adult patients with non-end-stage CKD develop CBF alterations in specific brain regions associated with cognitive impairment.

Therefore, the present study explored possible local CBF abnormalities in adult patients with stage 3–4 CKD by using the 3D-pCASL technique. We hypothesized that CBF alterations are already present in patients with stage 3–4 CKD compared to healthy controls and that these changes are associated with clinical biochemical indicators or cognitive impairment. 

## 2. Materials and Methods

### 2.1. Participants

All participants were recruited by the Guangdong Second Provincial General Hospital and gave informed consent. We used Tc-99m DTPA–based single-photon emission computed tomography to estimate the glomerular filtration rate (GFR) in patients with CKD while sorting recruited CKD patients according to the Kidney Disease Outcome Quality Initiative (K/DOQI) guidelines [16], defining a GFR of 30–59 mL/min/1.73 m^2^ as CKD stage 3 (CKD3) and a GFR of 15–29 mL/min/1.73 m^2^ as CKD stage 4 (CKD4); thus, 37 patients with CKD (18 patients in stage 3 and 19 patients in stage 4) were recruited for the study. Participants who were recruited for this study ranged in age from 20–65 years, and their renal function had remained constant for at least three months; patients who had undergone dialysis or kidney transplantation, which could affect outcomes, were excluded from the study. The exclusion criteria were severe neurological or psychiatric disorders, a history of drug or alcohol abuse, and a history of traumatic brain injury. In addition, 31 healthy adults of similar age, gender, and education level to the CKD participants were recruited as HCs. All participants in the study were right-handed. The flow chart of our whole study was shown in Figure 1.

### 2.2. Neuropsychological Tests

All participants completed a variety of neuropsychological tests prior to MRI data collection. The test scales included the Mini-Mental State Examination (MMSE), the Montreal Cognitive Assessment (MoCA), the number connection test A (NCT-A), and the digit symbol test (DST). The MoCA is an assessment tool used to quickly screen several areas of cognitive function, including episodic memory, language function, attention, orientation, delayed recall, visuospatial ability, and executive function. The MMSE is a widely used comprehensive assessment tool for cognitive function, and the NCT-A and DST mainly reflect attention and thought-tracking ability.

### 2.3. Clinical Laboratory Tests

Various clinical biochemical indicators (including hemoglobin, creatinine, blood urea nitrogen (BUN), serum potassium levels, and serum calcium levels) were performed in all patients with CKD prior to MRI data collection. No laboratory blood tests were carried out on HCs. All patients underwent blood sampling by professional staff, and laboratory tests for all patients were performed at the Department of Laboratory Science, Guangdong Second Provincial General Hospital.

### 2.4. Acquisition of MRI Data

All the CKD patients and HCs underwent MRI data acquisition. All MRI data were acquired using a Philips Ingenia 3.0 T scanner (Ingenia; Philips, Best, The Netherlands) with a 32-channel phased-array head coil. Structural brain data were acquired using a whole-brain 3D T1-weighted brain volume (BRAVO) imaging sequence with the following acquisition parameters: slice thickness/gap = 1.0/0 mm, number of sagittal slices = 185, flip angle = 8, repetition time (TR)/echo time (TE) = 7.8/3.6 ms, number of signal averages (NSA) = 1, matrix = 256 × 256, field of view (FOV) = 240 mm × 240 mm, scan time = 356 s. CBF data were acquired using a pCASL sequence with 3D fast spin-echo acquisition and background suppression for resting-state perfusion imaging with acquisition parameters of TR = 4155 ms; TE = 33 ms, number of excitations = 1, postlabeling delay = 2.64 s, number of layers = 20, layer thickness/layer spacing = 6/0 mm, FOV = 240 mm × 240 mm, and scan time = 291 s.

### 2.5. Processing of Cerebral Blood Flow Data

The pCASL images were analyzed on a Philips postprocessing workstation. All subjects’ CBF maps were spatially preprocessed using Statistical Parametric Mapping (SPM12, The Wellcome Centre for Human Neuroimaging, UCL Queen Square Institute of Neurology, London, UK, http://www.fil.ion.ucl.ac.uk/spm, accessed on 25 September 2022) software; the steps were as follows: (1) image alignment and spatial normalization: the individual 3D BRAVO structural images were aligned to the individual ASL images. The individual T1’ brain map was aligned to the standard-space T1 template to generate the alignment information from individual space to standard space, and the alignment deformation information was written to the individual CBF brain map to complete the alignment process from individual space to standard space CBF, thus generating a CBF map in standard space. (2) Z-transformation: This transformation was performed on the standardized data with the following equation: individual zCBF = (individual CBF-mean group CBF)/standard deviation group CBF. This normalization procedure reduces the influence of individual differences on the comparison between groups. (3) Spatial smoothing: The z-transformed CBF maps were smoothed using a Gaussian smoothing kernel with a full width at half maximum of 6 mm to improve the signal-to-noise ratio of the images.

### 2.6. Statistical Analysis

Two-sample *t*-tests were performed to evaluate pairwise differences in CBF between the CKD3, CKD4, and HC groups, with sex, age, and education as controlled covariates. A Gaussian random field theoretical correction for multiple comparisons at the cluster level was used to correct and identify brain regions with significantly different CBF between the three groups (with correction thresholds of *p* < 0.001 at the voxel level and *p* < 0.05 for applied clustering).

All statistical calculations were conducted in IBM SPSS software version 26.0 (Chicago, IL, USA). For demographic and clinical characteristics, the data of samples that satisfied normal distribution are presented as the mean ± standard deviation and were tested by independent-samples t tests or one-way ANOVA; the sex difference between groups was tested using χ^2^ tests. The abnormal distribution of sample data was presented as the median and interquartile range and was tested by the Kruskal—Wallis H test. We extracted the CBF values of significantly different brain regions and used partial correlation analysis to calculate the correlation coefficients between the CBF values and clinical indicators of CKD3 and CKD4 (hemoglobin, creatinine, BUN, serum potassium level, and serum calcium level) as well as neurological test scores (NCT-A, DST, MMSE, MoCA), with age and years of education as covariates. A value of *p* < 0.05 was considered statistically significant.

To investigate the ability of CBF values in differentiated brain regions to identify CKD and HCs, receiver operator characteristic curves (ROC) was constructed for the study. We calculated the Youden index and selected the cutoff values based on the maximum Youden index. *p* values < 0.05 were considered statistically significant.

## 3. Results

### 3.1. Demographic and Clinical Characteristics

The demographic and clinical characteristics of CKD3 patients, CKD4 patients, and HCs are shown in Table 1. The CKD3, CKD4, and control groups were found to be similar in age, sex, and years of education. The CKD4 group had significantly lower MMSE, MoCA, and DST scores than HC and CKD3 patients.

### 3.2. Differences in CBF between the Groups

The CKD4 group was found to have decreased CBF in the right insula and increased CBF in the left hippocampus compared to the HC group (Figure 2).

No brain regions were found to have significant CBF differences between the CKD3 and CKD4 groups or between the CKD3 and HC groups.

### 3.3. Correlation of Clinical Biochemical Indicators and Neuropsychological Scores with CBF among Subgroups

In the CKD4 sample, partial correlation analysis found that CBF in the right insula was negatively correlated with NCT-A scores (r = −0.544, *p* = 0.024); CBF in the left hippocampus was positively correlated with BUN (r = 0.649, *p* = 0.005) and negatively correlated with serum calcium (r = −0.646, *p* = 0.005) (Figure 3).

### 3.4. ROC Curves

As shown in Figure 4, the AUCs of the right insula and the left hippocampus indicate that both of these have the ability to identify CKD and HC. The right insula AUC was 0.861 (*p* < 0.01), and the left hippocampus AUC was 0.862 (*p* < 0.01); both brain regions had good accuracy in identifying patients with CKD. Table 2 shows that the cutoff value was 0.19 for the right insula (sensitivity, 63.20%; specificity, 93.50%) and −0.27 for the left hippocampus (sensitivity, 89.5%; specificity, 77.4%). The Yordon index, cutoff values, sensitivity, and specificity for the right insula and the left hippocampus are shown in Table 3. CBF values in the right insula and left hippocampus of CKD and HC, as well as the coordinate points of the ROC curves, the corresponding sensitivities, and 1-specifics are shown in the Appendix A.

## 4. Discussion

Based on 3D-pCASL, we found that the CKD4 group had decreased CBF in the right insula and increased CBF in the left hippocampus. The CKD4 group had significantly lower DST, MMSE, and MoCA scores than the HC group. Additionally, we found that the right insula CBF was negatively correlated with NCT-A, the left hippocampal CBF was positively correlated with BUN, and the left hippocampal CBF was negatively correlated with serum calcium. By comparing the AUC, it demonstrated that altered CBF values in the right insula and left hippocampus have good ability to identify CKD.

Our study found an increase in CBF in the left hippocampal region of CKD4 patients. The hippocampus plays an important role in cognitive memory and reasoning decisions [17,18]. Our results are consistent with those of previous studies [19,20]. For instance, Cheng BC et al. used ASL to study CBF changes in ESRD patients on long-term peritoneal dialysis therapy and compared CBF in ESRD patients before and after peritoneal dialysis with that of HCs; they found that ESRD patients had a mean increase in CBF compared to HCs and that higher CBF in the left hippocampus of ESRD patients was associated with worse executive function [19]. Li et al. studied CBF changes in elderly adults before and after hemodialysis and found that global CBF within the gray matter and white matter was significantly higher in hemodialysis patients than in HCs, with significant local CBF increases in the hippocampus and orbitofrontal cortex [20]. These studies found CBF alterations in the hippocampus as we did, but the difference is that our study was conducted in patients with CKD4, which also suggests that alterations in CBF are already present in non-end-stage patients. Additionally, we found a positive correlation between increased CBF values in the left hippocampus and BUN. Higher BUN levels were identified as a risk factor for the progression of renal disease in CKD patients [21], which further supports the possibility that hyperperfusion of CBF in the left hippocampus of CKD patients may be related to the severity of CKD.

Interestingly, decreased CBF in the right insula of CKD4 patients was also found in our results, and decreased CBF in the right insula of CKD4 patients was negatively correlated with NCT-A test scores. The insula is a cortical structure located deep in the brain between the frontal and temporal lobes and is thought to be closely related to emotional processing, attention, and cognitive control [22,23,24]. The results of our study showed a negative correlation between decreased CBF in the right insula and NCT-A test scores, suggesting that decreased CBF in the right insula may be closely related to the severity of cognitive impairment in the CKD4 stage. However, a previous study of non-end-stage CKD patients reported that children and young adults with CKD had higher global CBF than HCs [15]. This is not consistent with our results, and CBF changes in the insula were not found in this study. This contrasting result may be explained by the following possible reasons: First, the variability of the results may be related to the different ages of CKD patients. Previous studies suggest that brain metabolism and blood supply may play an important role in age-related cognitive decline and that CBF at the insular cortex, prefrontal cortex, and caudate is the area most affected by age [25]. Furthermore, the difference in outcome may be related to inadequate cerebral perfusion due to impaired renal function in CKD. Esposito et al. showed that perfusion changes occurring in renal ischemic injury could induce the appearance of molecules with profibrotic effects, leading to renal impairment [26]. Impaired or worsening renal function has been found to be associated with accelerated endothelial damage and the development of disease in small cerebral vessels, whose intact cerebral autoregulatory mechanisms may fail and put the brain at risk for inadequate perfusion [27,28,29]. Sedaghat S et al., in their study of the association between different levels of renal function and CBF, also found that CBF was lower in patients with poorer renal function [30]. Chronic cerebral hypoperfusion may cause neuronal damage, central cholinergic dysfunction, and oxidative damage in the brain and lead to cognitive impairment [31,32,33]. Therefore, we speculate that reduced CBF in the right insula leads to inadequate perfusion in the corresponding brain regions and thus may be one of the possible causes of cognitive impairment in CKD. Finally, the difference in outcomes may be related to the susceptibility of CKD to concurrent cerebrovascular and neurodegenerative diseases [34]. A systematic review and meta-analysis reported an increased risk of stroke in CKD patients with an estimated GFR <60 mL/min [35]. In a four-year follow-up survey of approximately 2000 CKD patients with chronic coronary artery disease, Nira Koren-Morag et al. found that slight renal insufficiency was strongly associated with an increased risk of ischemic stroke or transient ischemic attack in patients with preexisting atherosclerotic thrombotic disease [36]. Based on the above references, we speculate that the possibility of hypoperfusion of some brain regions and even microischemic stroke may already be present in patients with non-end-stage CKD.

Significantly, we also identified a negative correlation between increased CBF in the left hippocampus and serum calcium levels by bias correlation analysis, suggesting that hypocalcemia may influence cerebral hyperperfusion at some level in patients with CKD. Previous studies have generally accounted for cerebral hyperperfusion in CKD patients at rest as a compensatory dilation of small cerebral vessels secondary to hypertension and renal anemia [15,19,20,37]. The correlation between hypocalcemia and hypertension involves calcium hormones and blood pressure regulators and is multidimensional and complicated [38]. In a study investigating the clinical biochemical characteristics and the prevalence of bone mineral metabolism disorders in CKD patients at different stages without dialysis, it was found that as the severity of renal failure increased, serum calcium significantly decreased and hyperparathyroidism and arterial hypertension increased [39]; thus, we assume that low serum calcium levels in CKD patients during CKD progression may further contribute to the development of hypertension in such patients, which indirectly contributes to cerebral hyperperfusion. Previous studies have reported an association between cerebral hyperperfusion and cognitive impairment in CKD. Moreover, additional evidence suggests that disturbances in calcium homeostasis affect neurons, abnormal serum calcium levels are believed to be associated with depressive symptoms and cognitive impairment [40,41,42]; and hypocalcemia is known to directly stimulate the production of excess parathyroid hormone, which is thought to be neurotoxic [43,44]. These suggest that low calcium levels in CKD patients may be associated with cerebral hyperperfusion, but few studies are known on how calcium metabolism affects cognitive function in CKD patients, and more follow-up studies are still needed to verify it. However, in our study, we could not detect a correlation between increased CBF in the left hippocampus of CKD4 patients and neurocognitive test results, which may be influenced by the relatively small sample size.

Despite its strengths, our study still has some limitations. First, the small size of our study sample may compromise and constrain the reliability and generalizability of the results; for instance, we failed to observe meaningful results related to CKD3 patients. We will need to collect a larger number of cases and increase the age range of patients in the sample to validate our results. Second, this was a cross-sectional study. We did not determine whether the relationship between CBF changes and cognitive impairment changed with the progression of CKD disease, and a longitudinal study may help to address this question.

## 5. Conclusions

In conclusion, our study showed that CBF alterations were already present in patients with non-end-stage CKD and that the cognition-related brain regions, i.e., the left hippocampus and right insula, CBF alterations in patients with CKD stage 4, including increased CBF values in the left hippocampus of CKD stage 4 patients, may be related to the severity of CKD and reduced blood calcium concentration, and decreased CBF values in the right insula may be related to the severity of cognitive impairment. More notably, the altered CBF values in the right insula and left hippocampus had a good ability to identify CKD. These findings provide valuable insights for further understanding the relationship between altered cerebral perfusion and cognitive impairment in non-end-stage CKD patients, as well as identifying the possible neurophysiological mechanisms behind them.

## Figures and Tables

**Figure 1 jpm-13-00142-f001:**
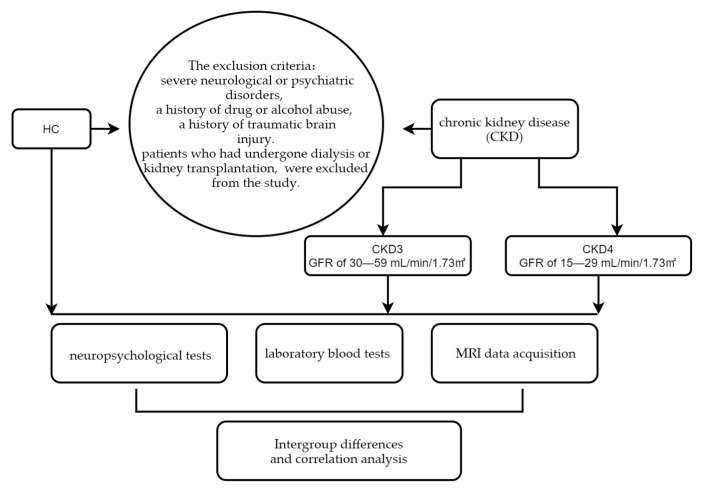
The study flow chart is depicted. HC, healthy controls.

**Figure 2 jpm-13-00142-f002:**
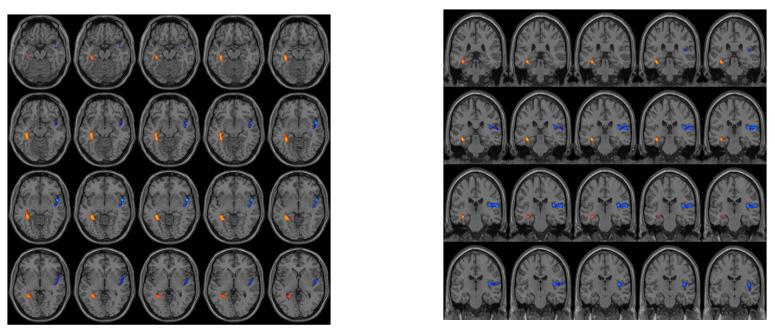
Differences in CBF between the groups. The CKD4 group was found to have decreased CBF in the right insula and increased CBF in the left hippocampus compared to the HC group.

**Figure 3 jpm-13-00142-f003:**
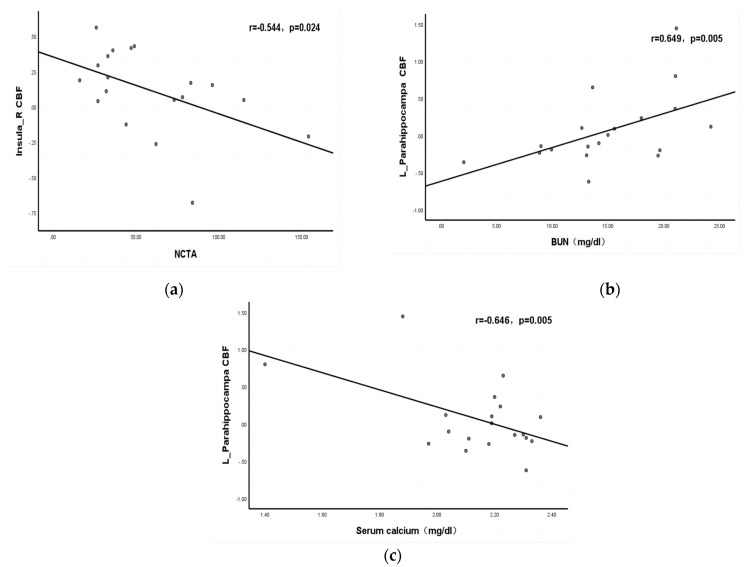
Correlation of clinical biochemical indicators and neuropsychological scores with CBF among subgroups (**a**). Partial correlation analysis found that CBF in the right insula was negatively correlated with NCT-A scores (r = –0.544, *p* = 0.024); (**b**) CBF in the left hippocampus was positively correlated with BUN (r = 0.649, *p* = 0.005); (**c**) CBF in the left hippocampus was negatively correlated with serum calcium (r = –0.646, *p* = 0.005).

**Figure 4 jpm-13-00142-f004:**
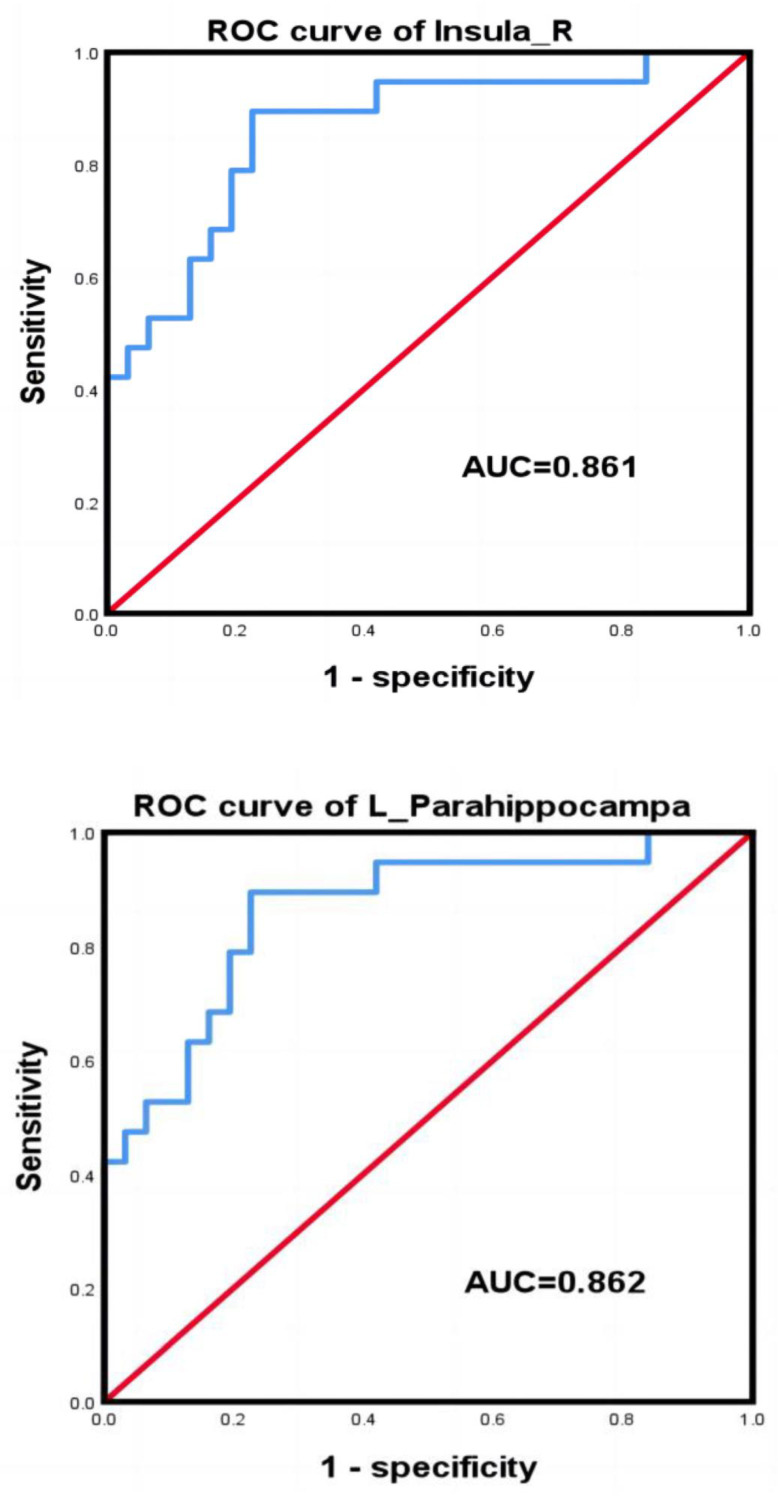
The ROC curves for the right insula and left hippocampus. The AUC for the right insula was 0.861, while the AUC for the left hippocampus was 0.862. Blue line: ROC curve of the right insula and left hippocampus; Red line, reference line.

**Table 1 jpm-13-00142-t001:** Demographics, neuropsychological tests, and laboratory results of CKD3, CKD4 patients, and HCs.

Clinical Measure	HC (*n* = 31)	CKD3 (*n* = 18)	CKD4 (*n* = 19)	*p*-Value	CKD3 vs. CKD4	HC vs. CKD3	HC vs. CKD4
Age (Y)	44.13 ± 10.63	44.61 ± 14.74	47.53 ± 11.35	0.609			
Sex (M/F)	18/13	13/5	11/8	0.579			
Education (Y)	9.71 ± 4.10	8.50 ± 3.03	8.00 ± 3.23	0.228			
Hemoglobin (g/dL)		114.72 ± 20.72	101.42 ± 17.61	0.042			
Urea nitrogen (mg/dL)		10.88 ± 3.43	14.98 ± 5.43	0.010			
Creatinine (mg/dL)		262.27 ± 100.31	437.95 ± 168.44	0.001			
Kalium (mg/dL)		4.12 ± 0.42	4.58 ± 0.54	0.007			
Calcium (mg/dL)		2.25 ± 0.25	2.14 ± 0.22	0.169			
eGFR (mL/min/1.73 m^2^)		39.00 ± 7.84	21.04 ± 4.50	<0.001			
MoCA score	27.00 (24.00,29.00)	25.00 (22.00,28.00)	24.00 (19.00,28.00)	0.024	1.000	0.209	0.031
MMSE score	29.00 (28.00,30.00)	29.00 (27.00,30.00)	28.00 (26.00,29.00)	0.020	0.281	1.000	0.016
NCT-A score	38.00 (33.00,47.00)	45.00 (37.00,71.00)	47.00 (32.00,83.00)	0.163			
DST score	45.00 (43.00,57.00)	40.50 (20.25,56.00)	34.00 (17.00,49.00)	0.014	1.000	0.201	0.015

MoCA, Montreal Cognitive Assessment; MMSE, Mini-Mental State Examination; NCT-A, type A number connection test; DST, digit symbol test.

**Table 2 jpm-13-00142-t002:** Brain regions showing abnormal CBF with the right insula and the left hippocampus in the CKD4 group (*p* < 0.05).

Brain Area (AAL)	VOXEL		Peak MNI Coordinate		Peak
right insula	257	50	6	−6	−5.0958
left hippocampus	70	−36	−40	−8	4.8948

MNI, Montreal Neurological Institute.

**Table 3 jpm-13-00142-t003:** Yordon index, cutoff values, sensitivity, and specificity for the right insula and the left hippocampus.

Brain Regions	Sensitivity	Specificity	Youden Index	Cut-Off
right insula	63.20%	93.50%	0.57	0.19
left hippocampus	89.5%	77.4%	0.67	−0.27

## Data Availability

The raw data supporting the conclusions of this article will be made available by the authors, without undue reservation.

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
