# Peer review of "Altered Cerebral Blood Flow in the Progression of Chronic Kidney Disease"

_jpm, 2023, doi:10.3390/jpm13010142_

Round 1

Reviewer 1 Report

Comments and Suggestions for Authors

The manuscript written by WeiZhao et al is an original article proposing to have a contribution to how chronic kidney disease (CKD) leads to cognitive impairment, mainly in patients with CKD stage 4. The manuscript is well written in terms of the English language. However, it requires improvement by reviewing a few major and minor issues:

Major:

  1. Lines 27-28 and line 218: it is a bold statement that “increased or decreased regional CBF may be associated with the progression of CKD disease”. Your research was a cross-sectional case-control study, not a follow-up study, so it cannot support the association with CKD progression.
  2. Lines 29-30: Following the same idea of the study design, causal relationships cannot be established by a cross-sectional case-control study. Therefore, the phrase “hypocalcemia may directly or indirectly affect cerebral perfusion alterations and cognitive function” is exaggerated.
  3. It does not appear anywhere that the healthy controls (HC) performed the imaging investigations (see chapter 2.4).
  4. Figure 1 is inconclusive: the patients with CKD3, CKD4 or HC are not identified, and there is no arrow that highlights what to focus the reader's attention on.
  5. Lines 187-189: The ROC curve was conducted without the numerical data for the CBF of the left hippocampus or right insula being presented.
  6. Please explain in the discussion section the role of the left hippocampus in cognitive impairment.
  7. Lines 316-317:  this wording is incomprehensible “both of the right insula and left hippocampus have a good ability and value for identifying CKD”. It goes without saying that only their existence (right insula and left hippocampus) could identify CKD patients !! Moreover, what is the added value of the fact that it can identify these patients? The identification of patients with CKD is clearly established according to the values of glomerular filtration, a reliable and cost-effective analysis. Why would I do a sophisticated imaging analysis to see that "both of the right insula and left hippocampus" (only their existence???) can identify patients with CKD? I don't think this is a goal in itself.

Minor:

  1. There are numerous abbreviations without being defined in advance (line 14 CBF, line 21 NCT-A,  line 91 ECT), or others that are used only once (line 57 PD and HD;  line 141 GRF)
  2. Line 18-19: If you performed Receiver operator characteristic (ROC), please explain for what reason.
  3. Line 39: Please replace “health problem” with “health issue”. It has a better scientific resonance.
  4. There are some typos: line 25 what is “to identify CKD with HC”; line 26 double “and”; line 97 MR abbreviation instead of MRI; etc
  5. Lines 68, 71, 76: What do “clinical biochemical indicators” mean? It must be specifically defined.
  6. Lines 85-86: There is a reference missing for the Kidney Disease Outcome 85 Quality Initiative (K/DOQI) guidelines.
  7. Line 182: it is not specified in whom this partial correlation was found (patients with CKD 4, CKD 3, or in all patients?)

After the amendment of the above comments in the manuscript, I would be in favor of publishing the article.

Author Response

请参考附件。

Reviewer 2 Report

Thank you for the opportunity  to  review  this manuscript.

Being  an integral  approach to the CKD patient of utmost importance, your research represents  a great opportunity area.

There  are  considerable points  to  take into  account:

1) Specify if you calculated your sample size.

2)Add a figure (flowchart)  on the  patient  recruitment, including  exclusion and  exclusion criteria.

3) Cerebral  blood flow  can  hardly be considered a  diagnostic  test  for CKD, so statistical  analysis  should be focused in the  relationship  of CBF with  neurological clinical  data, and trying  to  understand  if there  is  a relationship  between CBF and CKD progression. (i.e. chronologically parallel)

Author Response

请参阅附件。

Reviewer 3 Report

Lin WZ and coworkers evaluate the association of chenges in cerebral blood flow and cognitive impairment in patients with stage 3 and 4 chonic kidney disease. Cerebral blood flow is measured in different brain regions using MRI. A group of healthy subjects is used as a control group. The results demonstrate that subjects with chronic kidney disease stage 3 are affected by changes of brain perfusion. Furthermore, alterations in cerebral  perfusion can predict chronic kidney disease. The authors hypothesize the  maladaptive cerebral perfusion may lead to the changes causing the cognitive impairment characteristic of the chronic kidney disease patients. The study is of some interest because it explains the mechanisms underlying the cognitive alterations present in patients with chronic kidney disease. The authors demonstrate how some brain areas are hypoperfused while in others blood flow is increased. With a punctual acquisition of MRI images it is not possible to understand if the brain regions show perfusion variations over time. It has been demonstrated that the variation of perfusion as occurs in ischemia perfusion injury can induce the appearance of molecules with a profibrotic effect that lead to  chronic organ injury(Esposito et al. American Journal of Nephrology 2011, 33:239-249). The authors should speculate on this hypothesis in the discussion. Finally, while the association of early-stage CKD with changes in cerebral perfusion is important, it is perhaps not terribly interesting to be able to predict CKD based on cerebral perfusion. The measurement of the latter is certainly time-consuming and costly.

Author Response

请参阅附件。

Reviewer 4 Report

Grammar, sentence structure, and flow must be addressed throughout the entire manuscript. Also, the spacing between words needs to be addressed throughout the manuscript. 

The abstract is not grammatically formatted correctly. For example, take out the (1), (2), etc. 

The introduction needs more references after each sentence to support each statement. Also, there are inaccuracies in the background regarding CKD. 

1) The early stages of CKD may be reversible. However, the sentence suggests that renal health and filtration can be improved based on the CKD stage. 

2) CKD has been rising for decades, not the last decade.

The research question at the end of the introduction is too broad and needs to be narrowed down. Also, the research question/questions need to be stated without explaining the methods in the introduction.

Methods

How and where were the chemical laboratory tests performed?

Statistics

What is the sample size based on a power analysis?

Results

The spacing in Table 1 must be spread out more between the values (ex: SD symbols and parenthesis).

Figures 3 and table 2 are right on top of each other and need to be spaced out. 

Discussion, the first paragraph needs to be more direct. The authors start expounding on the results of the study before they finish highlighting the main findings. 

Lines 282-284 need to be rewritten to provide clarity better. 

In the limitation section, what is meant by CKD3 in line 308?
